# A Body of Circumstantial Evidence for the Irreversible Ectonucleotidase Inhibitory Action of FSCPX, an Agent Known as a Selective Irreversible A_1_ Adenosine Receptor Antagonist So Far

**DOI:** 10.3390/ijms22189831

**Published:** 2021-09-11

**Authors:** Gabor Viczjan, Tamas Erdei, Ignac Ovari, Nora Lampe, Reka Szekeres, Mariann Bombicz, Barbara Takacs, Anna Szilagyi, Judit Zsuga, Zoltan Szilvassy, Bela Juhasz, Rudolf Gesztelyi

**Affiliations:** 1Department of Pharmacology and Pharmacotherapy, Faculty of Medicine, University of Debrecen, H-4032 Debrecen, Hungary; viczjan.gabor@pharm.unideb.hu (G.V.); erdei.tamas@pharm.unideb.hu (T.E.); ovari.ignac@gmail.com (I.O.); lampenori@gmail.com (N.L.); rszekeres57@gmail.com (R.S.); bombicz.mariann@pharm.unideb.hu (M.B.); takacs.barbara@pharm.unideb.hu (B.T.); dr.szilagyi.anna@med.unideb.hu (A.S.); szilvassy.zoltan@med.unideb.hu (Z.S.); juhasz.bela@med.unideb.hu (B.J.); 2Doctoral School of Nutrition and Food Sciences, University of Debrecen, H-4032 Debrecen, Hungary; 3Department of Health Systems Management and Quality Management for Health Care, Faculty of Public Health, University of Debrecen, H-4032 Debrecen, Hungary; zsuga.judit@med.unideb.hu

**Keywords:** extracellular adenosine, ectonucleotidase, POM-1, PSB-12379, FSCPX, NBMPR, NBTI, A_1_ adenosine receptor, heart, atrium, rat, guinea pig

## Abstract

In previous studies using isolated, paced guinea pig left atria, we observed that FSCPX, known as a selective A_1_ adenosine receptor antagonist, paradoxically increased the direct negative inotropic response to A_1_ adenosine receptor agonists (determined using concentration/effect (E/c) curves) if NBTI, a nucleoside transport inhibitor, was present. Based on mathematical modeling, we hypothesized that FSCPX blunted the cardiac interstitial adenosine accumulation in response to nucleoside transport blockade, probably by inhibiting CD39 and/or CD73, which are the two main enzymes of the interstitial adenosine production in the heart. The goal of the present study was to test this hypothesis. In vitro CD39 and CD73 inhibitor assays were carried out; furthermore, E/c curves were constructed in isolated, paced rat and guinea pig left atria using adenosine, CHA and CPA (two A_1_ adenosine receptor agonists), FSCPX, NBTI and NBMPR (two nucleoside transport inhibitors), and PSB-12379 (a CD73 inhibitor), measuring the contractile force. We found that FSCPX did not show any inhibitory effect during the in vitro enzyme assays. However, we successfully reproduced the paradox effect of FSCPX in the rat model, mimicked the “paradox” effect of FSCPX with PSB-12379, and demonstrated the lipophilia of FSCPX, which could explain the negative outcome of inhibitor assays with CD39 and CD73 dissolved in a water-based solution. Taken together, these three pieces of indirect evidence are strong enough to indicate that FSCPX possesses an additional action besides the A_1_ adenosine receptor antagonism, which action may be the inhibition of an ectonucleotidase. Incidentally, we found that POM-1 inhibited CD73, in addition to CD39.

## 1. Introduction

The group of cardiovascular diseases has been killing the most people in the world. At a global level, the leading cause of death was ischemic heart disease, which was responsible for 16% of the world’s total deaths in 2019, and, since 2000, produced the largest increase in deaths [1]. Remarkably, such a rate of cardiovascular mortality is uncommon among non-human mammals. In adult wild and domestic mammals, heart disease generally accounts for less than 11% of deaths, most of which can be traced back to conditions unrelated to ischemia (e.g., parasitic infections or valvular diseases). This indicates the existence of human-specific factors in the pathogenesis of ischemic heart disease, which probably include physical inactivity and chronic psychic stress [2]. These observations underline the importance of research efforts aiming an understanding of heart physiology and pathophysiology, with special emphasis on interspecies differences. An important topic in this field is the investigation of endogenous cardioprotective mechanisms such as the cardiac adenosinergic system [3,4,5,6], a system in which species dependence is also present [7]. Of course, in the usual sequence of information acquisition in life sciences, animal experiments will precede human studies for a long time to come.

Adenosine is a purine nucleoside that plays a special role in the nucleic acid metabolism by modulating a variety of physiological and especially pathological processes throughout the body. Accordingly, adenosine is a substrate for several enzymes and transporters [8,9]. CHA (*N*^6^-cyclohexyladenosine) and CPA (*N*^6^-cyclopentyladenosine) are synthetic adenosine analogues that act as selective, biologically stable (enzyme-resistant, especially CHA [10]) and high-efficacy (especially CPA [11]) full agonists for the A_1_ adenosine receptor (A_1_ receptor). The A_1_ receptor is the predominant adenosine receptor type of the myocardium that mediates several protective processes including negative tropic effects (primarily on the supraventricular myocardium) [8,12,13]. In the isolated, paced left atrium, the direct (i.e., detectable without a previous enhancement of the contractile force) negative inotropic effect, characteristic of the atrial muscle [8,12], can serve as a well-measurable and reliable output of the A_1_ receptor function [11].

Nitrobenzylthioinosine derivatives (e.g., NBMPR and NBTI, see Section 4.2.1) are selective inhibitors of the nucleoside transporter type ENT1 (a.k.a. SLC29A1), which is a major equilibrative adenosine carrier in the heart [14,15,16]. Nitrobenzylthioinosines substantially modify the tissue distribution of molecules that are transported by ENT1 to a significant extent. Such molecules are usually highly metabolized in certain tissue compartments, mostly in the cell interior, causing inward transport for such compounds [17,18]. Consistently, nitrobenzylthioinosines can drastically transform the concentration–effect (E/c) curve of adenosine receptor agonists through two mechanisms that we can distinguish as a general and a specific one [19]. The general modifying effect of nitrobenzylthioinosines affects the E/c curve of all adenosine receptor agonists because it is mediated by a change in the interstitial level of endogenous adenosine resulted from the blockade of the transmembrane adenosine flux. In the heart, ENT1 blockade usually increases the interstitial concentration of endogenous adenosine, since, under physiological conditions, the adenosine flux is directed into the cell interion [17]. This general influence of nitrobenzylthioinosines is manifested in a decrease in E_max_ and increase in EC_50_ (for explanation of these abbreviations, see Equation (1) in Section 4.2.4) (Figure 1, left panel) [19]. The other “specific” effect of nitrobenzylthioinosines only influences the E/c curve of adenosine receptor agonists that are highly transported via ENT1 (e.g., adenosine itself). The specific effect stems from an increase in the interstitial concentration of such agonists, when administered during the construction of an E/c curve, due to the inhibition of ENT1 (which could remove these agonists from the microenvironment of their cell-surface receptors). If the specific effect can prevail alone, it increases E_max_ and diminishes EC_50_, but usually, the resultant effect of the general and specific influences is detected (Figure 1, right panel) [19].

FSCPX (see Section 4.1.1) is known as a selective, irreversible A_1_ receptor antagonist [11,20,21,22]. However, in earlier studies, we observed that a pretreatment with FSCPX seemed to paradoxically increase the direct negative inotropic response to adenosine and CPA, but only when NBTI was also present in the system (Figure 1) [19,23,24]. To solve this “FSCPX paradox”, we hypothesized that FSCPX pretreatment inhibited the general, but not the specific, effect of NBTI, probably by blunting the interstitial adenosine production [19]. Until the writing of the present work, this hypothesis has been based on ex vivo data obtained from one single animal model (isolated and paced guinea pig left atrium) with the use of adenosine, CPA, FSCPX, and NBTI (as illustrated by the Figure 1). To properly evaluate these data and to test their interpretation, a simple but highly specialized mathematical model was previously developed [25,26] and then applied [27,28]. As a mechanism of FSCPX paradox, we supposed that FSCPX might inhibit one (or some) enzyme(s) participating in the interstitial formation of adenosine [19,28]. This supposition enables that FSCPX only interferes with the general modifying effect of NBTI, which affects the interstitial level of endogenous, but not exogenous, adenosine.

Differentiation between endogenous and exogenous adenosine is justified by the fact that when analyzing E/c curves in a traditional manner, an increase in the interstitial concentration of the endogenous adenosine has the opposite effect on the E/c curve of an adenosine receptor agonist as an increase in the interstitial level of the exogenous adenosine has on its own E/c curve [19,23,24]. Nitrobenzylthioinosines, by preventing adenosine from intracellular elimination, elevate the interstitial level of adenosine, irrespective of its origin [17]. However, nitrobenzylthioinosines, being administered before the E/c curve is constructed, accumulate the endogenous adenosine even before generating the E/c curve, so this surplus adenosine can consume a portion of the response capacity of A_1_ receptors and thereby reduce the detectable effect of an exogenous A_1_ receptor agonist (added later for the E/c curve) (Figure 1, left panel). Contrary to this, nitrobenzylthioinosines protect the exogenous adenosine exclusively during the construction of the E/c curve, so the effect of the preserved extra exogenous adenosine is added to the resultant effect, increasing that (Figure 1, right panel). It should be noted that most of the synthetic A_1_ receptor agonists (including CHA and CPA), as compared to adenosine, are greatly resistant to adenosine-handling enzymes [10]; therefore, their concentration is only slightly affected by nitrobenzylthioinosines (in the time window of our experiments). Consequently, the biologically stable synthetic agonists are suitable for examining exclusively the general E/c curve-modifying effect of nitrobenzylthioinosines (Figure 1, left panel), while exogenous adenosine enables the investigation of the specific E/c curve-modifying effect (occurring usually together with the general one) (Figure 1, right panel).

In the present study, our goal was to test our hypothesis about the inhibitory effect of FSCPX on one (or a few) adenosine-forming enzyme(s) in the interstitium. It was an obvious strategy to first investigate the possible involvement of ectonucleotidases in the FSCPX paradox. Ectonucleotidases play a key role in the extracellular generation of adenosine (Table 1). In the heart, the two most important ectonucleotidases are CD39 (ecto-apyrase) and CD73 (ecto-5′-nucleotidase), which together catalyze three consecutive steps: decomposition of ATP, via ADP, to AMP (plus two inorganic orthophosphate ions) by CD39, and furthermore, the breakdown of AMP to adenosine (and one inorganic orthophosphate ion) by CD73 [4,29]. First, we carried out enzyme inhibitor assays aiming to directly explore the effect of FSCPX on CD39 and/or CD73. Next, we continued the investigation with various indirect approaches to get closer to answering the question.

## 2. Results

### 2.1. CD39 and CD73 Inhibitor Assays

DMSO, present in the wells even in 10% (*v*/*v*), did not significantly influence the activity of CD39 and CD73. POM-1, an inhibitor of E-NTPDase1-3 enzymes [30] (Table 1) and antagonist of some P2 purinergic receptors [31], significantly but incompletely reduced the activity of CD39. Interestingly, POM-1 also inhibited CD73, at least to the same extent as it inhibited CD39. (Starting from the fact that the level of inhibition barely increased by increasing the POM-1 concentration from 20 to 200 μM, this could be the maximal inhibitory effect of POM-1 for both enzymes in these assays.) In turn, PSB-12379, an inhibitor recently developed for CD73 [32,33], significantly decreased the activity of CD73, which action was selective (did not cover to CD39) and complete at 1 μM. However, neither FSCPX nor CPX appeared to significantly affect the activity of either CD39 or CD73, despite the high concentrations administered (Figure 2).

### 2.2. Interaction of FSCPX with NBMPR in the Rat Left Atrium

#### 2.2.1. Response to Adenosine before Any In Vitro Treatment

Adenosine, the non-selective, physiological adenosine receptor agonist, concentration-dependently decreased the resting contractile force of rat left atria in all groups used for this investigation (see Figure 3, left panel). Responses to adenosine did not differ significantly, at any concentrations, among the groups (handled uniformly at this stage of the experiments). This result denotes the homogeneity of the groups regarding the main adenosinergic mechanisms (Figure 3, left panel).

#### 2.2.2. Response to CHA in the Control Groups

CHA, a selective, synthetic A_1_ receptor full agonist, also reduced the resting contractile force of rat left atria in a concentration-dependent manner. Responses to CHA did not differ significantly between the two control groups, indicating that the difference in the duration of their protocols did not significantly influence the effect of CHA on the atria (Figure 3, right panel). (As the protocols using FSCPX were considerably longer than the other ones, two control protocols, a long (Control for FSCPX and FSCPX+NBMPR) and a short one (Control for NBMPR) were elaborated for the proper comparison with protocols using and not using FSCPX.)

#### 2.2.3. Response to CHA after Different In Vitro Treatments

When administered alone, both FSCPX and NBMPR blunted the effect of CHA in the rat left atrium in a statistically significant manner: FSCPX pushed the CHA E/c curve considerably to the right without reducing the maximal response, while NBMPR moderately decreased the response to CHA (seemingly including the maximal response). However, a pretreatment with FSCPX made the effect of NBMPR on the response to CHA negligible: the CHA E/c curves in the FSCPX and FSCPX+NBMPR groups practically coincided (Figure 4).

These results provided by rat atria corroborate the results of our previous investigation using guinea pig atria [19] (presented by the Figure 1). Although sole effects of FSCPX and nitrobenzylthioinosines on the response to the synthetic A_1_ receptor agonists, regarding their magnitude (but not direction), exhibit considerable differences between the rat and guinea pig models, it is clear that FSCPX pretreatment reduced the effect of nitrobenzylthioinosines in atria of both species (cf. Figure 1, left panel and Figure 4).

### 2.3. Interaction of PSB-12379 with NBTI in the Guinea Pig Left Atrium

#### 2.3.1. Response to Adenosine in All Groups

Regarding the first adenosine E/c curves of the groups (before any in vitro treatment), adenosine concentration-dependently decreased the resting contractile force of guinea pig left atria. The response to adenosine did not differ significantly among the groups at any concentrations, proving the homogeneity of the groups in terms of the adenosinergic mechanisms (Figure 5, left panel).

Similarly, when compared the third adenosine E/c curves of groups Control+NBTI+PSB (for Ado) and Control+PSB+NBTI (for Ado), responses to adenosine did not differ significantly at any concentrations. Thus, the previous treatments (with NBTI or with PSB-12379) did not disturb the outcome of the final co-treatment with PSB-12379 and NBTI (Figure 5, right panel). (In the absence of prior knowledge, it was unclear whether a previous NBTI or PSB-12379 treatment could influence a subsequent NBTI and PSB-12379 co-treatment. Thus, the co-treatment with NBTI and PSB-12379 was performed after both NBTI and PSB-12379 treatments. Accordingly, these groups were named Control+NBTI+PSB (for Ado) and Control+PSB+NBTI (for Ado), respectively.)

#### 2.3.2. Response to CPA in the Groups Labeled with “for CPA”

CPA, a selective, synthetic A_1_ receptor full agonist, also decreased the resting contractile force of guinea pig left atria in a concentration-dependent manner. The effect of NBTI was similar to that of NBMPR in the rat atrium and practically the same as our earlier results with NBTI in the guinea pig atrium (cf. Figure 6, left panel, Figure 1 and Figure 4, left panel). When added alone, PSB-12379, a selective inhibitor of CD73 with no apparent effect on the A_1_ receptor, did not significantly influence the effect of CPA, in contrast to FSCPX. However, PSB-12379 significantly blunted the effect of NBTI on the response to CPA, just as FSCPX did. The only difference between the “anti-NBTI” effects of PSB-12379 and FSCPX was that the former was weaker than the latter (cf. Figure 6, left panel and Figure 1, left panel).

#### 2.3.3. Response to Adenosine in the Groups Labeled with “for Ado”

The effect of NBTI was practically the same as that observed in our earlier work (cf. Figure 6, right panel and Figure 1, right panel). Similar to the results with CPA (Figure 6, left panel), PSB-12379, administered alone, did not significantly affect the effect of adenosine (Figure 6, right panel). Importantly, also consistent with the results with CPA, PSB-12379 significantly inhibited the effect of NBTI for the response to adenosine; more precisely, it inhibited the so-called general E/c curve-modifying effect of NBTI (Figure 6). Thus, PSB-12379, when co-administered with NBTI, moderately but statistically significantly increased the response to adenosine as compared to the adenosine E/c curve generated in the presence of NBTI alone (Figure 6, right panel). Additionally, in line with the results with CPA, this effect of PSB-12379 was weaker than that of FSCPX (cf. Figure 6, right panel and Figure 1, right panel).

In summary, these results demonstrate that PSB-12379, which restricted the interstitial adenosine formation by blocking its final step, produced the same kind of transformation in the CPA and adenosine E/c curves generated in the presence of NBTI (Figure 6) as the pretreatment with FSCPX did in these curves, of course in addition to the consequences of A_1_ receptor blockade (Figure 1). This finding has provided indirect evidence for an inhibitory effect of FSCPX, a verified irreversible A_1_ receptor antagonist, on the interstitial adenosine production in the heart (or, at least, in the rat and guinea pig left atrium). Starting from our experimental conditions, this additional effect of FSCPX appeared also to be irreversible.

### 2.4. The Influence of Different Administration Regimens on the Effect of FSCPX in Rat and Guinea Pig Left Atria

#### 2.4.1. Response to Adenosine before Any In Vitro Treatment

Adenosine concentration-dependently diminished the resting contractile force of the left atria, whose diminution was greater for guinea pigs than for rats (from 10 μM adenosine concentration). The effect of adenosine did not differ significantly, at any of the concentrations, among the groups related to the same species. As the different groups received the same treatment until this stage of these experiments, this result indicates the homogeneity of the groups, within the same species, regarding the adenosinergic mechanisms (Figure 7). In addition, a greater sensitivity of the (Hartley type) guinea pig atrium to adenosine (in terms of the direct negative inotropy) was demonstrated as compared to the (Wistar-type) rat atrium (Figure 7).

#### 2.4.2. Response to CPA after an In Vitro Pretreatment with 10 μM FSCPX Using Different Administration Regimens

As expected, CPA reduced the resting contractile force of atria in a concentration-dependent manner. The response of guinea pig left atria to CPA was spectacularly greater than that of rat ones. The FSCPX pretreatment (exposure of atria to 10 μM FSCPX for 45 min followed by 75 min wash-out) produced a substantial dextral displacement of the CPA E/c curves with no diminution of the maximal effect (in comparison with the DMSO-pretreated control CPA E/c curves), in both species (Figure 8). However, the magnitude of this antagonistic effect exerted by FSCPX statistically significantly depended on the number of administrations of FSCPX to the bathing medium of atria. When we attempted to maintain 10 μM concentration of FSCPX for 45 min via one single addition of FSCPX (dissolved in DMSO), the antagonistic effect was the weakest, upon both species, in comparison with cases when FSCPX was administered more than once (not exceeding a concentration of 10 μM in the organ baths at any time) (Figure 8).

Regarding the rat atria, the increase in the number of administrations of FSCPX from two to five did not enhance the irreversible antagonism elicited by FSCPX. In contrast, in the guinea pig atria, when FSCPX was administered five times (of course not exceeding the 10 μM concentration), the antagonism was the strongest among the FSCPX-pretreated samples (Figure 8). Importantly, even quintuple administration of FSCPX (“FSCPX regimen with five cycles”) was not able to reduce the maximal effect of CPA in both species (Figure 8).

For the rat atria, fitting of the CPA E/c curve data (averaged within the groups) to the Hill model (Equation (1)) provided moderately different E_max_ and Hill coefficient (n) values with substantially different logEC_50_ values (when computing the antilog, EC_50_ values range almost two orders of magnitude) (Table 2, panel “R-i”). Accordingly, fitting of the CPA E/c curves of guinea pig atria to Equation (1) also yielded similar E_max_ and Hill coefficient values and considerably different logEC_50_ values (ranging about one and a half orders of magnitude at the level of EC_50_ values) (Table 2, panel “GP-i”). This pattern broadly meets the appearance of competitive antagonism. As FSCPX is a verified irreversible antagonist for the A_1_ receptor, this result is indicative of a great A_1_ receptor reserve for the direct negative inotropic effect of CPA in the atrial myocardium of both species. Starting from this observation, the fitting of Equation (1) was repeated with shared E_max_ and n parameters to illustrate the influence of the different FSCPX administration regimens (panels “R-g” and “GP-g” of Table 2, Figure 8).

Regarding the rat atria, the effect of FSCPX could only be enhanced by increasing the number of FSCPX administrations to two, probably due to reaching the maximal effect of FSCPX in this system (Table 2, panels “R-i” and “R-g”). In turn, with regard to the guinea pig atria, the more cycles in the FSCPX administration regimen of a group that existed, the bigger the logEC_50_ value of the given group was (Table 2, panels “GP-i” and “GP-g”). These observations are consistent with the location of the CPA E/c curves of the different FSCPX-pretreated groups (Figure 8).

## 3. Discussion

By providing a body of indirect evidence, the results of the present study have corroborated our earlier hypothesis that FSCPX has an additional action besides the A_1_ adenosine receptor antagonism, through which it decreases the interstitial adenosine level. Furthermore, FSCPX seems to exert its effects in the cell membrane, in which it may accumulate after administration, while its fraction remaining dissolved in a water-based solution rapidly decomposes. As CD39 (ecto-apyrase) and CD73 (ecto-5′-nucleotidase) enzymes are membrane-bound (just like the A_1_ receptor), CD39 and CD73 have remained the most likely possible targets for the additional action of FSCPX. Incidentally, we have found that POM-1, thought to be an inhibitor of E-NTPDase1-3 enzymes (including E-NTPDase1 *a.k.a.* CD39) and the antagonist of some P2 purinergic receptors, inhibits CD73 as well.

The key to understanding the present study is understanding the location of E/c curves, displayed in Figure 1, relative to one another. Our ex vivo results are based on evaluating E/c curve transformations, from which conclusions have been drawn regarding the underlying mechanisms. In Figure 1, FSCPX pushed the E/c curve of both CPA and adenosine to the right (in comparison with the appropriate Control E/c curves) that reflects the loss of A_1_ receptors, which is consistent with the irreversible A_1_ receptor antagonist nature of FSCPX [11,23,24]. The retained maximal effect of the FSCPX-pretreated E/c curves implies a great A_1_ receptor reserve for the effect measured (direct negative inotropy) [11,23,24], but it does not matter much for the present study. NBTI (the nitrobenzylthioinosine derivative used) pushed the E/c curve of CPA to the right and decreased the maximal effect. Since nitrobenzylthioinosines do not antagonize the A_1_ receptor [14,16], the above-mentioned phenomenon indicates an effect that diminished the response of the A_1_ receptors (expressed by the E/c curves) in some other manner. This is the so-called general E/c curve-modifying effect of NBTI that has been attributed to the interstitially accumulated endogenous adenosine produced by the inhibition of the inward adenosine transport elicited by NBTI [34,35]. The effect of NBTI on the adenosine E/c curve is somewhat more complex, because it results from two components: the general E/c curve-modifying effect (see above) and a specific one, which enhances the A_1_ receptor response by increasing the amount of exogenous adenosine (via inhibition of adenosine transport that prevents exogenous adenosine from the intracellular elimination). If the effects of FSCPX and NBTI were simply added together, the E/c curves reflecting the co-action would be shifted to the right from the NBTI-treated E/c curves, but this is not the case. Instead, the FSCPX+NBTI-treated E/c curves show a sinistral displacement from the NBTI-treated E/c curves and exceed them (irrespective of the agonist used), indicating a significant interaction between effects of FSCPX and NBTI. Based on ex vivo and in silico pieces of evidence, this interaction was finally attributed to a previously unknown effect of FSCPX, by which it can reduce the general E/c curve-modifying effect of NBTI [19,28]. The possible mechanism of this additional effect of FSCPX is the main topic of the present study.

Accordingly, the essence of our ex vivo investigations was a reverberation between the shape of E/c curves and the molecular causes determining it: interpreting an E/c curve transformation based on the known mechanisms of action of agents used and, conversely, assuming a mechanism of action from E/c curve transformations observed. During this process, we built up our final conclusion from the acquired knowledge step by step.

In earlier studies dealing with E/c curves generated in isolated and paced guinea pig left atria, we found that FSCPX, a molecule widely known as a selective and irreversible A_1_ receptor antagonist, paradoxically increased the maximal response to adenosine and CPA, A_1_ receptor agonists, in the presence of NBTI, which is a selective inhibitor of the nucleoside transporter type ENT1 [23,24]. Subsequently, in the background of this paradoxical phenomenon, an interference was suggested between effects of FSCPX and NBTI, based on results of an in silico modeling [27]. During the further ex vivo and in silico investigations of this FSCPX paradox, it was proposed that that FSCPX, in addition to antagonizing the A_1_ receptor, inhibits one of the two effects of NBTI, namely the interstitial accumulation of endogenous (but not exogenous) adenosine produced by ENT1 blockade [19,28]. As a mechanism to do this, it was hypothesized that FSCPX might blunt the activity of one or some interstitial adenosine-forming enzyme(s), most likely CD39 and/or CD73, which are the two main ectonucleotidases in the heart [4,8,29]. The aim of the present study was to verify or reject this hypothesis, preferably by providing direct evidence for or against the enzyme inhibitory activity of FSCPX.

In the present study, direct evidence for the ectonucleotidase inhibitory action of FSCPX would have been the inhibition of CD39 and/or CD73 by FSCPX during a reliable in vitro inhibitor assay. However, when using kits containing the enzymes and their verified and putative inhibitors in an aqueous solution, neither CD39 nor CD73 were inhibited by FSCPX (Figure 2). The cause of this outcome can be the fast degradation of FSCPX (in 2–3 min) in aqueous solutions [36]. Additionally, CPX (see Section 4.1.1), a selective and reversible antagonist of the A_1_ receptor, the initial structure for the synthesis of FSCPX, also failed to inhibit CD39 and CD73 (Figure 2). As CPX, apart from its poor water solubility, seems to be sufficiently stable in water [37], it can be concluded that this part of FSCPX itself is certainly not responsible for any ectonucleotidase inhibitory action.

To resolve the contradiction between the ex vivo efficiency and in vitro inefficiency of FSCPX, we have assumed that FSCPX, after administration, rapidly enters the lipid compartment of the tissues (probably mostly cell membranes), while its fraction remaining dissolved in aqueous solutions (bathing medium, interstitial fluid, cytosol) quickly decomposes. Thus, FSCPX might act on structures in/on the lipid compartment, such as membrane-associated proteins. Indeed, the A_1_ receptor, CD39, and CD73 are all membrane-bound proteins [8,29]. If this assumption is true, FSCPX cannot be examined with water-based kits.

At this point of the study, we returned to the ex vivo approach. First, we aimed to confirm the original phenomenon (FSCPX paradox), so we repeated the most reliable part of the original experiment (where a synthetic A_1_ receptor agonist was applied; see Figure 1, left panel) with some modifications: instead of CPA, NBTI, and guinea pig, we used CHA, NBMPR and rat, respectively. These modifications served the purpose of reducing the influence of accidental properties of the previously used animal model and agents on the outcome. We found that although the location of the rat CHA E/c curves considerably differed from the corresponding guinea pig CPA E/c curves (showing a stronger effect of FSCPX and a weaker action of NBMPR in the rat atrium), the phenomenon of FSCPX paradox appeared in the rat model as well. To make this clear, the two nitrobenzylthioinosine-treated E/c curves (without and with an FSCPX pretreatment in the same animal model) had to be compared to their real controls (in terms of the size of the operable A_1_ receptor population). Thus, the solely NBMPR- and solely NBTI-treated E/c curves were contrasted with the appropriate Control E/c curves, whereas the FSCPX+NBMPR- and FSCPX+NBTI-treated E/c curves with the appropriate solely FSCPX-treated E/c curves (of course, within the same species). This comparison showed that the FSCPX pretreatment drastically decreased the effect of the given nitrobenzylthioinosine derivative on the E/c curve of the given synthetic A_1_ receptor agonist (cf. Figure 1 and Figure 4, left panel). As this effect of NBMPR and NBTI is exclusively the so-called general E/c curve-modifying effect that is mediated by an increase in the interstitial concentration of endogenous adenosine, it can be concluded that the FSCPX pretreatment can reduce the interstitial adenosine formation. Thus, the results of this investigation (about the interaction of FSCPX with NBMPR in the rat atrium) corroborated our previous results displaying the FSCPX paradox (illustrated by the Figure 1).

As the next step, we aimed to investigate whether there is an interaction between an agent, which has been proven to restrict the interstitial adenosine production and is unable to antagonize the A_1_ receptor, and a nitrobenzylthioinosine derivative, which blocks the nucleoside transport, in the guinea pig atrium. Thus, this time, we repeated our whole original experiment (see Figure 1) with the replacement of FSCPX with PSB-12379, which is a recently developed potent CD73 inhibitor [32,33]. We found that the presence of PSB-12379 transformed the CPA and adenosine E/c curves the same way as the FSCPX pretreatment did, apart from the dextral displacement of E/c curves caused by the A_1_ receptor antagonist action of FSCPX that did not appear in response to PSB-12379. Namely, the general E/c curve-modifying effect of NBTI, which manifested in a decrease in E_max_ and increase in EC_50_, prevailed to a less extent when PSB-12379 was also present (cf. Figure 1 and Figure 6). Interestingly, the pretreatment with 10 μM FSCPX had visibly a stronger influence on the NBTI-treated E/c curves than the presence of 3 μM PSB-12379, although even 1 μM PSB-12379 exerted a maximal CD73 inhibitory effect in our in vitro investigation (Figure 2, right panel). It may be speculated that the highly lipid-soluble FSCPX can more efficiently inhibit the membrane-bound CD73 than the highly water-soluble PSB-12379. In summary, restriction of the interstitial adenosine formation mimicked the effect of FSCPX other than A_1_ receptor antagonism.

Finally, we aimed to get evidence to support our assumption regarding the kinetic aspects of actions of FSCPX. It has long been recognized that the action of a drug in the body is also a kinetic event in terms of both the reach to its target and the interaction with that [38]. So, we started from the idea that only the fraction of FSCPX that enters the cell membrane within minutes after administration can exert an effect. In this case, the more times FSCPX is administered during the 45-min incubation period (reaching the same concentration, 10 μM, in the bathing medium), the greater the amount (and concentration) of FSCPX in the cell membranes will be (of course, up to a certain limit). For this purpose, we investigated the influence of an increasing number of administrations (and wash-out periods) on the efficiency of FSCPX in terms of A_1_ receptor antagonism in the rat and guinea pig atria. We found that the effect of FSCPX could be enhanced by increasing the number of FSCPX administrations until two (in rat) and until five, which is the applied maximum (in guinea pig) (Figure 8, Table 2). This outcome supports our assumption on the priority role of the cell membrane in the development of actions of FSCPX and thereby provides a possible resolution for the contradiction between the ex vivo efficiency and in vitro inefficiency of FSCPX in terms of the inhibitory action on CD39 and/or CD73. This kinetic property of FSCPX explains why the mechanisms of action of this agent are difficult to explore.

Instead of one overriding piece of direct evidence, we have collected three pieces of indirect evidence for the additional activity of FSCPX, by which it inhibits the interstitial adenosine-accumulating effect of nucleoside transport blockers. Declaring that clarification of the exact molecular mechanism of the action of FSCPX in addition to the irreversible A_1_ receptor antagonism warrants further investigations, we propose that FSCPX has the ability to modify a membrane-bound target, as a result of which the interstitial adenosine production in the myocardium (or, at least, in the supraventricular one) narrows. Starting from the long (75-min) wash-out period after the FSCPX pretreatment, both effects of FSCPX appear to be irreversible, although the persistence of some FSCPX in the membranes after the wash-out cannot be excluded. The cause, because of which this additional effect of FSCPX is only revealed upon nucleoside transport blockade, is that even the resting level of endogenous adenosine in the interstitium is too low to evoke a significant negative inotropic effect, not to mention any decrease in it [39]. Therefore, only maneuvers significantly increasing the interstitial concentration of endogenous adenosine (e.g., nucleoside transport blockade) can manifest the additional action of FSCPX mentioned above. Nevertheless, currently, our experimental approach is the only ex vivo system that has enabled the exposure and further investigation of the FSCPX paradox.

## 4. Materials and Methods

### 4.1. In Vitro Enzyme Inhibitor Assays

#### 4.1.1. Materials

Applied chemicals and kits: 8-cyclopentyl-1,3-dipropylxanthine (CPX or DPCPX), 8-cyclopentyl-*N*^3^-[3-(4-(fluorosulfonyl)benzoyloxy)propyl]-*N*^1^-propylxanthine (FSCPX), sodium polyoxotungstate (POM-1; as a part of the CD39 Inhibitor Screening Assay Kit, see below), disodium *N*^6^-benzyl-α,β-methyleneadenosine-5′-diphosphate (PSB-12379), CD39 Inhibitor Screening Assay Kit, and CD73 Inhibitor Screening Assay Kit.

CPX and DMSO were purchased from Merck KGaA (Darmstadt, Germany). FSCPX was manufactured by Santa Cruz Biotechnology, Inc. (Heidelberg, Germany) and distributed by BIO-Kasztel, Ltd. (Budapest, Hungary). Kits (including POM-1) were produced by BPS Bioscience (San Diego, CA, USA) and distributed by THP Medical Products Vertriebs GMBH (Vienna, Austria). PSB-12379 was manufactured by Tocris Bioscience (Bristol, UK) and distributed by Bio-Techne R&D Systems, Ltd. (Budapest, Hungary). Redistilled water was used to dissolve and dilute POM-1 as well as dilute the pre-dissolved PSB-12379. FSCPX and CPX were dissolved in dimethyl-sulfoxide (DMSO; purchased from Merck KGaA (Darmstadt, Germany)), and then, they were diluted with redistilled water (when needed).

#### 4.1.2. Protocol

The level of inhibition of CD39 (ecto-apyrase) and CD73 (ecto-5′-nucleotidase) elicited by POM-1, PSB-12379, FSCPX, and CPX was determined by means of the malachite green colorimetric method, which was compliant with the instructions of the manufacturer of assay kits used. For both kits, 14 sorts of mixtures were prepared for the 96-well microplates: negative control (blank; in 8 wells), negative control with 1% DMSO (in 4 wells), negative control with 10% DMSO (in 4 wells), positive control (in 8 wells), positive control with 1% DMSO (in 4 wells), positive control with 10% DMSO (in 4 wells), inhibitor control with 20 μM POM-1 (in 8 wells), inhibitor control with 200 μM POM-1 (in 8 wells), inhibitor control with 0.1 μM PSB-12379 (in 8 wells), inhibitor control with 1 μM PSB-12379 (in 8 wells), inhibitor test of 10 μM FSCPX (in 1% DMSO; in 8 wells), inhibitor test of 100 μM FSCPX (in 10% DMSO; in 8 wells), inhibitor test of 10 μM CPX (in 1% DMSO; in 8 wells), and inhibitor test of 100 μM CPX (in 10% DMSO; in 8 wells). To quantify the inorganic orthophosphate produced by CD39 and CD73 (and thereby to characterize enzyme activities), the absorbance at 630 nm was measured with Varioskan LUX Multimode Microplate Reader obtained from Thermo Fisher Scientific (Waltham, MA, USA).

### 4.2. Ex Vivo Functional Assays

#### 4.2.1. Materials

As a bathing medium for all preparations, modified Krebs–Henseleit buffer (Krebs solution) was used that contained (in mM): NaCl: 118, KCl: 4.7, CaCl_2_: 2.5, NaH_2_PO_4_: 1, MgCl_2_: 1.2, NaHCO_3_: 24.9, glucose: 11.5, and ascorbic acid: 0.1, dissolved in redistilled water.

The following agents were used: adenosine, *N*^6^-cyclohexyladenosine (CHA), *N*^6^-cyclopentyladenosine (CPA), *S*-(4-nitrobenzyl)-6-thioinosine (NBMPR), *S*-(2-hydroxy-5-nitrobenzyl)-6-thioinosine (NBTI), FSCPX, and PSB-12379 (see Section 4.1.1). Although the abbreviations “NBMPR” and “NBTI” are often used as synonyms, herein, they refer to different structures.

Agents were purchased from Merck KGaA (Darmstadt, Germany), except for FSCPX and PSB-12379 (see above). Adenosine was dissolved in the 36 °C Krebs solution. CHA and CPA were dissolved in ethanol/water (1:4) solution (*v*/*v*). NBMPR, NBTI, and FSCPX were dissolved in DMSO. All these stock solutions were adjusted to a concentration of 10 mM. Normal saline (0.9% *w*/*v* of NaCl) was used to dilute the pre-dissolved PSB-12379 (adjusting it to 1 mM concentration). Adenosine, CHA, and CPA stock solutions were diluted with Krebs solution.

#### 4.2.2. Animals and Groups

The animal use protocols were approved by the Committee of Animal Research, University of Debrecen, Hungary (5/2020/DEMÁB). Male Wistar rats and male Hartley guinea pigs weighing 400–500 g and 500–700 g, respectively, were used. The animals were guillotined; then, the left atria were quickly removed and mounted at 10 mN resting tension in 10 mL vertical organ chambers (Experimetria TSZ-04; Experimetria Kft, Budapest, Hungary) filled with Krebs solution, aerated with 95% O_2_ and 5% CO_2_ (36 °C; pH = 7.4). Left atria were paced by platinum electrodes (3 Hz, 1 ms, twice the threshold voltage) by means of a programmable stimulator (Experimetria ST-02; Experimetria Kft, Budapest, Hungary) and power amplifier (Experimetria PST-02; Experimetria Kft, Budapest, Hungary). The contractile force was characterized by the amplitude of isometric twitches, which were measured by a transducer (Experimetria SD-01; Experimetria Kft, Budapest, Hungary) and strain gauge (Experimetria SG-01D; Experimetria Kft, Budapest, Hungary) and recorded by a polygraph (Medicor R-61 6CH Recorder; Medicor Művek, Budapest, Hungary).

When investigating the potential interaction of FSCPX and NBMPR (see Section 2.2), the rat left atria were randomly divided into five groups: Control for NBMPR group (n = 4), NBMPR group (n = 5), Control for FSCPX & FSCPX+NBMPR group (n = 5), FSCPX group (n = 5), and FSCPX+NBMPR group (n = 4).

To explore a potential interaction between PSB-12379 and NBTI (see Section 2.3), the guinea pig left atria were randomly divided into six groups: Control (for CPA) group (n = 4), NBTI (for CPA) group (n = 6), PSB (for CPA) group (n = 4), PSB+NBTI (for CPA) group (n = 5), Control+NBTI+PSB (for Ado) group (n = 7), and Control+PSB+NBTI (for Ado) group (n = 6).

During the investigation of the different FSCPX administration regimens (see Section 2.4), the rat and guinea pig left atria were randomized into four groups: DMSO (5 cycles) group (n = 10 and 7), FSCPX (1 cycle) group (n = 8 and 6), FSCPX (2 cycles) group (n = 9 and 7), and FSCPX (5 cycles) group (n = 8 and 8) (respectively).

#### 4.2.3. Protocols

In the organ chambers, all atria (isolated from both species) were first allowed to equilibrate in Krebs solution for 25 min; next, they were subjected to 100 μM adenosine for 2 min (as a priming). Afterwards, a wash-out was made with Krebs solution for 20 min. Next, a cumulative E/c curve was constructed with adenosine, which was followed by another 20-min long wash-out period (using Krebs solution). From this stage of the experiments, the different protocols continued in different ways.

When investigating the potential interaction between FSCPX and NBMPR (see Section 2.2), each rat left atrium underwent one of five protocols, which were the same as those that were used in a previous study, namely P1a, P1b, P2a, P2b, and P2c [19], except for three modifications. Herein, CHA was applied instead of CPA; NBMPR was used instead of NBTI; and some protocols performed in [19] were omitted from this investigation (where adenosine was used as a main agonist). Specifically, each atrium underwent an in vitro treatment as follows: the Control for the NBMPR group received 10 μL DMSO for 15 min; the NBMPR group was subjected to 10 μM NBMPR (administered with 10 μL DMSO) for 15 min; the Control for the FSCPX and FSCPX+NBMPR groups received 10 μL DMSO for 45 min followed by a 60-min wash-out, after which 10 μL DMSO was administered for 15 min; the FSCPX group received 10 μM FSCPX (with 10 μL DMSO) for 45 min followed by a 60-min wash-out, and then 10 μL DMSO was added for 15 min; the FSCPX+NBMPR group received 10 μM FSCPX (with 10 μL DMSO) for 45 min followed by a 60-min wash-out, after which 10 μM NBMPR (with 10 μL DMSO) was administered for 15 min. Finally, without washing out the last administered agent (and/or solvent), a cumulative CHA E/c curve was generated in all rat atria. It is of importance (see below) that the 45-min long incubation period was interrupted, at 22–23 min, with a short (half-min long) but intense wash-out, at the end of which the 10 μM FSCPX (with 10 μL DMSO) or the 10 μL DMSO alone was readministered. Thus, the administration of FSCPX to the atria in this investigation followed an “FSCPX (2 cycles)” regimen (similar to every previous FSCPX administration regimen used in our laboratory, until the present study).

When investigating the potential interaction between PSB-12379 and NBTI (see Section 2.3), each guinea pig left atrium was subjected to one of six protocols, similar to those described in the previous paragraph. The main differences are as follows: return to NBTI instead of NBMPR; return to adenosine and CPA instead of CHA; use of protocols, where adenosine was the main agonist, again (just like in [19]); and use of PSB-12379 to replace FSCPX (with modifications of wash-out and incubation periods consistent with this latter interchange). So, the Control (for CPA) group received 10 μL DMSO for 15 min; the NBTI (for CPA) group received 10 μM NBTI (added with 10 μL DMSO) for 15 min; the PSB (for CPA) group received 10 μL DMSO and 3 μM PSB-12379 for 15 min; the PSB+NBTI (for CPA) group received 3 μM PSB-12379 for 15 min and (without a wash-out) 10 μM NBTI (with 10 μL DMSO) for a further 15 min; the Control+NBTI+PSB (for Ado) group received 10 μM NBTI (with 10 μL DMSO) for 15 min; and the Control+PSB+NBTI (for Ado) group received 10 μL DMSO and 3 μM PSB-12379 for 15 min. Next, for groups labeled with “for CPA”, a cumulative CPA E/c curve was generated (and their protocols ended there), whereas for groups with “for Ado” in their names, a cumulative adenosine E/c curve was constructed (the second one for these atria). Afterwards, both groups labeled with “for Ado” received 3 μM PSB-12379 for 15 min and then 10 μM NBTI (with 10 μL DMSO) for a further 15 min. Finally, a third cumulative adenosine E/c curve was generated in these two groups. Thus, overall, the Control+NBTI+PSB (for Ado) and Control+PSB+NBTI (for Ado) groups provided four kinds of adenosine E/c curves: a control one (the first curve in both groups), an NBTI- and a PSB-12379-treated one (second curves of the groups, respectively), and a PSB+NBTI-treated one (the third curve in both groups).

When studying the different FSCPX administration regimens (see Section 2.4), each rat and guinea pig left atrium was subjected to one of four protocols. The DMSO (5 cycles) group received 10 μL DMSO for 45 min followed by a 75-min wash-out, while the other three groups were exposed to 10 μM FSCPX (administered with 10 μL DMSO) for 45 min followed by a 75-min wash-out. Importantly, the 45-min long incubation period was not interrupted in the FSCPX (1 cycle) group, whereas, in every other group, it was interrupted with one or more short but intense wash-out period(s) and subsequent readministration(s) of 10 μM FSCPX (with 10 μL DMSO) or 10 μL DMSO alone. In the FSCPX (2 cycles) group, the 45-min incubation was discontinued once (at 22–23 min) by making a wash-out and the readdition of 10 μM FSCPX (with 10 μL DMSO). In the FSCPX (5 cycles) group and DMSO (5 cycles) group, it was interrupted four times (every 8–9 min) with wash-out and readministration of 10 μM FSCPX (with 10 μL DMSO) and 10 μL DMSO alone, respectively. Finally, a cumulative CPA E/c curve was constructed in all atria.

#### 4.2.4. Characterization of the E/c Curves

The CPA E/c curves generated during the investigation of FSCPX administration regimens (see Section 2.4) were averaged within the groups and then fitted to the Hill equation [40]:E = E_max_ · c^n^/(c^n^ + EC_50_^n^),(1)
where E: the effect defined as a percentage decrease in the initial contractile force; c: the concentration of the agonist administered; E_max_: the maximal effect; EC_50_: the agonist concentration producing a half-maximal effect (midpoint location); n: the Hill coefficient (slope factor).

The curve fitting was performed two ways: individually (the averaged CPA E/c curves were fitted independently from each other) and globally (the averaged CPA E/c curves were fitted at once with shared E_max_ and n parameters).

#### 4.2.5. Data Analysis

Each atrium was required to meet three criteria in order to qualify for inclusion in the further evaluation: (i) the resting contractile force had to reach 1 mN before the adenosine E/c curve; (ii) the mechanical activity of the paced atrium had to be regular; (iii) the response to 10 μM adenosine was required to be within the mean ± 2 SD range (if more than one adenosine E/c curve occurred, this requirement applied to the first one). The mean and SD were computed using atria meeting the first two criteria.

Gaussian distribution of data and homogeneity of variances were tested with the Shapiro–Wilk test and Brown–Forsythe test, respectively. If all datasets (more than two) showed Gaussian distribution and homogenous variances, they were compared with ordinary one-way ANOVA followed by Tukey post-testing. Upon Gaussian distribution without the homogeneity of variances (even for one dataset), Welch’s one-way ANOVA with Dunnett’s T3 post-testing was performed. Without a Gaussian distribution (even for one dataset), Kruskal–Wallis test with Dunn’s post-testing was used.

Concentrations (c, EC_50_) in the Hill equation (Equation (1)) were expressed as common logarithms (logc, logEC_50_), as recommended [41]. Statistical analysis and curve fitting were carried out with GraphPad Prism 8.4.3 (686) for Windows (GraphPad Software Inc., La Jolla, CA, USA), while other calculations were made with Microsoft Excel 2016 (Microsoft Co., Redmond, WA, USA).

## 5. Conclusions

In previous ex vivo studies, we observed a paradoxical phenomenon; i.e., FSCPX, an agent widely known as a selective A_1_ receptor antagonist, increased the response to A_1_ receptor agonists (determined via evaluating E/c curves), if a nucleoside transport inhibitor was present in the system [19,23,24]. Based on the results of in silico investigations, we hypothesized that FSCPX blunted the cardiac interstitial adenosine accumulation caused by nucleoside transport blockade [27], in the background of which it might be that FSCPX inhibited CD39 and/or CD73 [28], which are the two main enzymes of the interstitial adenosine production in the heart [8,29]. In the present study, a body of indirect evidence was gathered to support our hypothesis. In our case, direct evidence would have been the demonstration of inhibition of CD39 and/or CD73 by FSCPX in an in vitro enzyme inhibitor assay that, however, failed. At the same time, we have demonstrated (i) the reproducibility of the original phenomenon in another animal model with a different A_1_ receptor agonist and nucleoside transport inhibitor; (ii) the imitability of the original phenomenon using a CD73 inhibitor lacking the A_1_ receptor antagonist property; (iii) the lipophilia of FSCPX that, on one hand, may be the reason for the negative outcome of the CD39 and CD73 inhibitor assays (a reason other than the lack of enzyme inhibitory activity of FSCPX), and, on the other hand, makes CD39 and CD73, two membrane-bound enzymes, the most likely targets for FSCPX to reduce the interstitial adenosine formation. According to the classic rope analogy (from 1866) in jurisprudence, each piece of circumstantial evidence is similar to a strand of a rope: strands on their own may be weak, but together they can be strong enough [42]. Taken together, the three pieces of indirect evidence mentioned above are strong enough to indicate that FSCPX possesses an additional action besides the A_1_ adenosine receptor antagonism, which action may be (at least according to our expectations) the inhibition of a membrane-bound ectonucleotidase. Accordingly, the next step of investigations may be to find the appropriate assay system, where the activity of the membrane-associated forms of CD39 and CD73 can be measured (and in which the preserved membranes can serve as a safe reservoir for FSCPX). Nevertheless, whatever new mechanism of action for FSCPX will be discovered, it may open new opportunities for the research of the adenosine-related mechanisms.

## Figures and Tables

**Figure 1 ijms-22-09831-f001:**
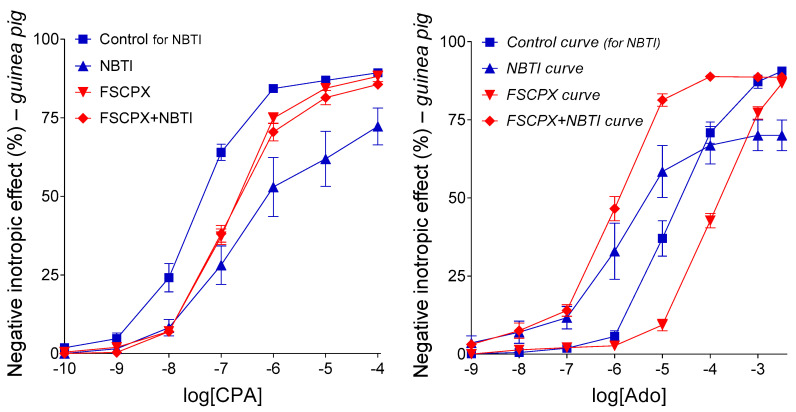
The direct negative inotropic effect of CPA (**left panel**), a synthetic full agonist of the A_1_ adenosine receptor with a relatively long half-life [10], and adenosine (**right panel**), the physiological adenosine receptor full agonist with a short half-life [10] in the isolated guinea pig left atrium, with and without a pretreatment with FSCPX combined with the absence and presence of NBTI. The *x*-axis shows the common logarithm of the molar concentration of the given agonist (in the bathing medium), while the *y*-axis denotes the effect (determined as a percentage decrease in the initial contractile force). The symbols indicate the responses to the agonists averaged within the groups (±SEM). The blue and red E/c curves refer to the absence and presence of an FSCPX pretreatment, respectively. Group names are written with upright letters, while different E/c curves obtained from the same groups are indicated in italics. CPA: *N*^6^-cyclopentyladenosine; Ado: adenosine; FSCPX: 8-cyclopentyl-*N*^3^-[3-(4-(fluorosulfonyl)benzoyloxy)propyl]-*N*^1^-propylxanthine; NBTI: *S*-(2-hydroxy-5-nitrobenzyl)-6-thioinosine. Data are redrawn from [19].

**Figure 2 ijms-22-09831-f002:**
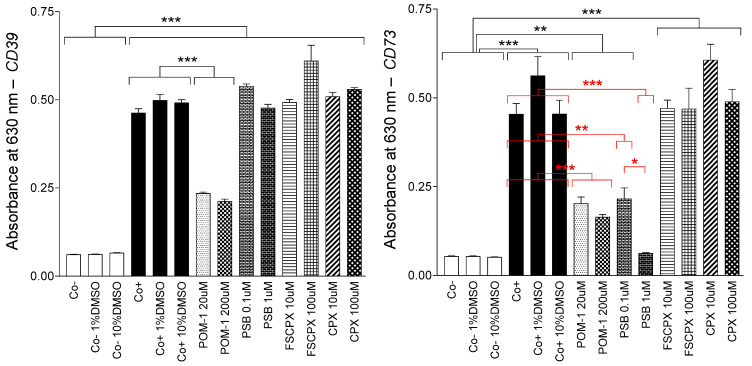
Absorbance at 630 nm measured in two 96-well microplates to quantify the activity of CD39 (**left panel**) and CD73 (**right panel**) enzymes (+ SEM). For both kits, 14 sorts of mixtures were prepared (each one in four or eight wells). Co-: negative control (“blank”); Co- 1%DMSO: negative control containing 1% DMSO; Co- 10%DMSO: negative control containing 10% DMSO; Co+: positive control; Co+ 1%DMSO: positive control containing 1% DMSO; Co+ 10%DMSO: positive control containing 10% DMSO; POM-1 20 μM: inhibitor control with 20 μM POM-1; POM-1 200 μM: inhibitor control with 200 μM POM-1; PSB 0.1 μM: inhibitor control with 0.1 μM PSB-12379; PSB 1 μM: inhibitor control with 1 μM PSB-12379; FSCPX 10 μM: inhibitor test of 10 μM FSCPX; FSCPX 100 μM: inhibitor test of 100 μM FSCPX; CPX 10 μM: inhibitor test of 10 μM CPX; CPX 100 μM: inhibitor test of 100 μM CPX; *: comparison of the sorts of mixtures indicated; the number of marks means the level of statistical significance (one mark: *p* < 0.05, two marks: *p* < 0.01, three marks: *p* < 0.001).

**Figure 3 ijms-22-09831-f003:**
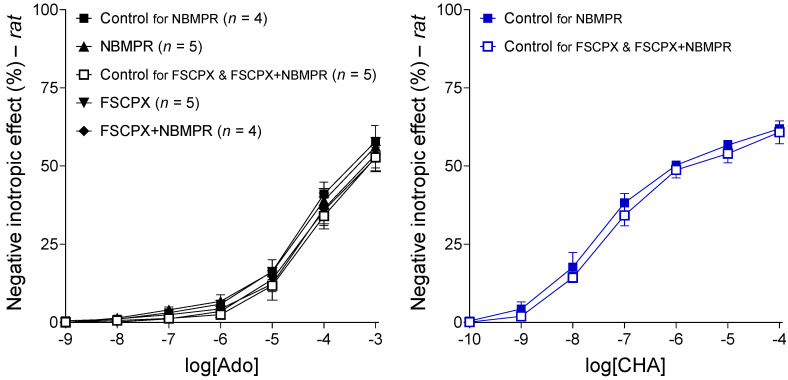
The direct negative inotropic effect of adenosine (**left panel**) and CHA (**right panel**) in the isolated rat left atrium, in the absence of any chemicals affecting the adenosinergic homeostasis. The left panel shows all groups that were handled uniformly until this stage of the investigation (the group names refer to subsequent in vitro treatments). The right panel presents the two control groups that underwent two in vitro treatments different in their duration. The *x*-axis denotes the common logarithm of the molar concentration of the given agonist (in the bathing medium), and the *y*-axis shows the effect (as a percentage decrease in the initial contractile force). The symbols indicate the responses to the agonists averaged within the groups (± SEM). The blue color used for the E/c curves in the right panel corresponds to the color code seen in Figure 4 (blue color refers to the absence of an FSCPX pretreatment). Ado: adenosine; CHA: *N*^6^-cyclohexyladenosine; E/c: concentration–effect.

**Figure 4 ijms-22-09831-f004:**
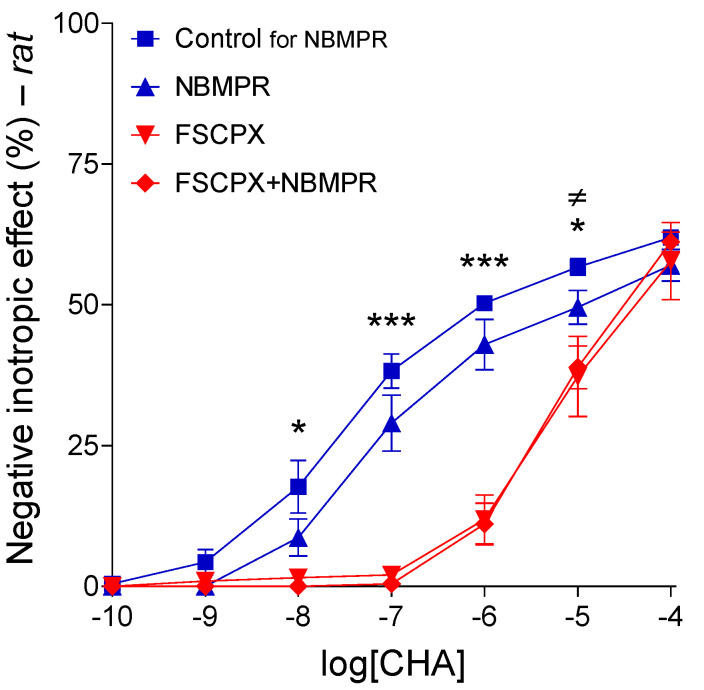
The direct negative inotropic effect of CHA in the isolated rat left atrium, without and with a pretreatment with 10 μM FSCPX (administered two times, severed by a short wash-out), combined with the absence and presence of 10 μM NBMPR. For simplicity, the Control for the FSCPX and FSCPX+NBMPR groups (not differing significantly from the Control for the NBMPR group) has been omitted. The *x*-axis denotes the common logarithm of the molar concentration of CHA (in the bathing medium), and the *y*-axis indicates the effect (as a percentage decrease in the initial contractile force). The symbols show the responses to the agonists averaged within the groups (± SEM). The blue and red E/c curves refer to the absence and presence of an FSCPX pretreatment, respectively. FSCPX: 8-cyclopentyl-*N*^3^-[3-(4-(fluorosulfonyl)benzoyloxy)propyl]-*N*^1^-propylxanthine; NBMPR: *S*-(4-nitrobenzyl)-6-thioinosine; CHA: *N*^6^-cyclohexyladenosine; E/c: concentration–effect; *: comparison of the Control for the FSCPX and FSCPX+NBMPR groups (data not shown) with the FSCPX group as well as FSCPX+NBMPR group (indicated on the CHA E/c curve of the Control for the NBMPR group); ≠: comparison of the Control for the NBMPR group with the NBMPR group; the number of marks means the level of statistical significance (one mark: *p* < 0.05, three marks: *p* < 0.001).

**Figure 5 ijms-22-09831-f005:**
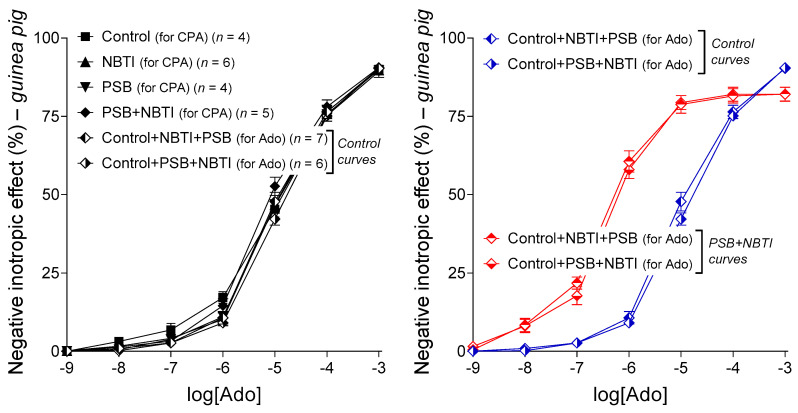
The direct negative inotropic effect of adenosine in the isolated guinea pig left atrium, in the absence of any chemicals affecting the adenosinergic homeostasis (for all groups) and in the presence of PSB-12379 together with NBTI (for the groups labeled with “for Ado”). The left panel denotes the first adenosine E/c curve of all groups (for the groups labeled with “for CPA”, this is the only adenosine E/c curve, while in the case of the groups labeled with “for Ado”, this is the so-called control curve). Atria producing the first adenosine E/c curve were handled uniformly, so the group names refer to subsequent in vitro treatments. The right panel presents the first and third adenosine E/c curves of the groups labeled with “for Ado” separately. The *x*-axis shows the common logarithm of the molar concentration of adenosine (in the bathing medium), and the *y*-axis denotes the effect (as a percentage decrease in the initial contractile force). The symbols indicate the responses to adenosine averaged within the groups (± SEM). The color code used for the E/c curves in the right panel is the same as seen in Figure 6 (blue and red colors refer to the absence and presence of PSB-12379, respectively). Kind of E/c curves provided by groups labeled with “for Ado” is indicated in italics. E/c: concentration–effect; Ado: adenosine; CPA: *N*^6^-cyclopentyladenosine; NBTI: *S*-(2-hydroxy-5-nitrobenzyl)-6-thioinosine; PSB-12379: disodium *N*^6^-benzyl-α,β-methyleneadenosine-5′-diphosphate.

**Figure 6 ijms-22-09831-f006:**
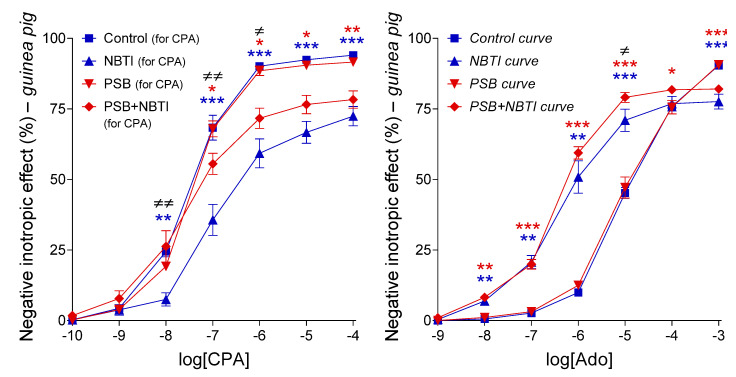
The direct negative inotropic effect of CPA (**left panel**) and adenosine (**right panel**) in the isolated guinea pig left atrium, in the absence and presence of 10 μM NBTI and 3 μM PSB-12379 (in all combinations). Since neither the two control curves nor the two PSB+NBTI curves differed significantly from each other (see the right panel of Figure 5), these curve pairs were pooled (as seen in the right panel here). The *x*-axis shows the common logarithm of the molar concentration of the given agonist (in the bathing medium), and the *y*-axis indicates the effect (as a percentage decrease in the initial contractile force). The symbols indicate the responses to the agonists averaged within the groups (± SEM). The blue and red E/c curves refer to the absence and presence of PSB-12379, respectively. The kind of E/c curves provided by groups labeled with “for Ado” is indicated in italics. E/c: concentration–effect; Ado: adenosine; CPA: *N*^6^-cyclopentyladenosine; PSB-12379: disodium *N*^6^-benzyl-α,β-methyleneadenosine-5′-diphosphate; NBTI: *S*-(2-hydroxy-5-nitrobenzyl)-6-thioinosine; blue asterisks: comparison of the Control (for CPA) group with the NBTI (for CPA) group (left panel), or comparison of the pooled control curve with the NBTI-treated curve (right panel); red asterisks: comparison of the Control (for CPA) group with the PSB+NBTI (for CPA) group (left panel) or comparison of the pooled control curve with the pooled PSB+NBTI-treated curve (right panel); ≠: comparison of the NBTI (for CPA) group with the PSB+NBTI (for CPA) group (**left panel**), or comparison of the NBTI-treated curve with the pooled PSB+NBTI-treated curve (**right panel**); the number of marks means the level of statistical significance (one mark: *p* < 0.05, two marks: *p* < 0.01, three marks: *p* < 0.001).

**Figure 7 ijms-22-09831-f007:**
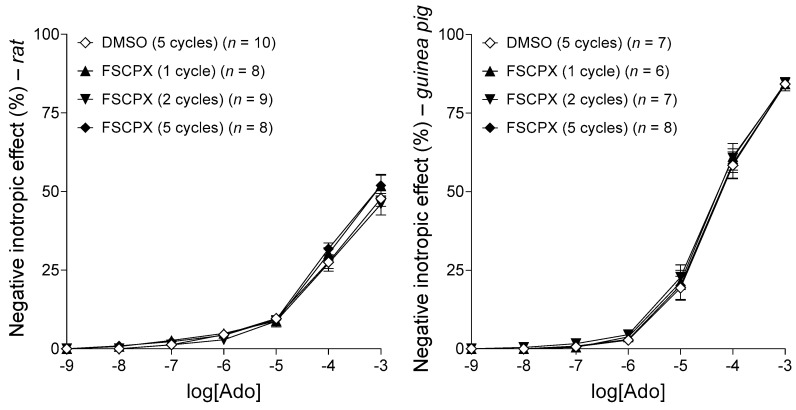
The direct negative inotropic effect of adenosine in the isolated rat (**left panel**) and guinea pig (**right panel**) left atrium, divided into four groups, handled uniformly at this stage of the investigation (thus, the group names refer to subsequent in vitro treatments). The *x*-axis shows the common logarithm of the molar concentration of adenosine (in the bathing medium), and the *y*-axis denotes the effect (as a percentage decrease in the initial contractile force). The symbols indicate the responses to adenosine averaged within the groups (± SEM). Ado: adenosine; FSCPX: 8-cyclopentyl-*N*^3^-[3-(4-(fluorosulfonyl)benzoyloxy)propyl]-*N*^1^-propylxanthine; DMSO: dimethyl-sulfoxide.

**Figure 8 ijms-22-09831-f008:**
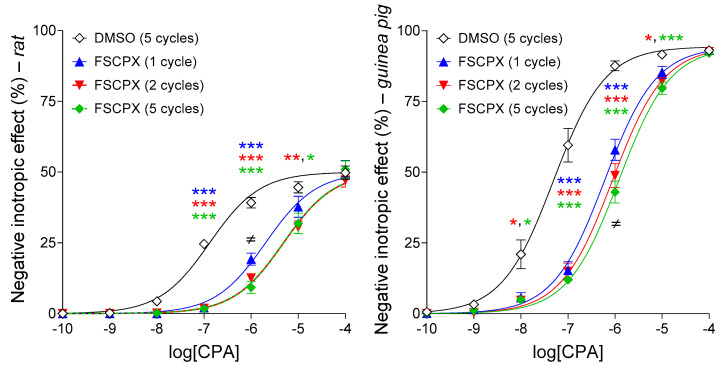
The direct negative inotropic effect of CPA in the isolated rat (**left panel**) and guinea pig (**right panel**) left atrium, with and without a pretreatment with 10 μM of FSCPX (carried out three different ways). The *x*-axis denotes the common logarithm of the molar concentration of CPA (in the bathing medium), and the *y*-axis shows the effect (as a percentage decrease in the initial contractile force). The symbols indicate the responses to CPA averaged within the groups (± SEM), while the lines show the globally fitted Equation (1) (with shared E_max_ and n parameters; see Table 2). The blue, red, and green E/c curves represent FSCPX pretreatments with one, two, and five administration(s) of FSCPX (each one preceded by a short wash-out), respectively, while the black E/c curve indicates the absence of an FSCPX pretreatment (but the presence of five administrations of DMSO, the solvent of FSCPX, preceded by five short wash-out periods). E/c: concentration–effect; FSCPX: 8-cyclopentyl-*N*^3^-[3-(4-(fluorosulfonyl)benzoyloxy)propyl]-*N*^1^-propylxanthine; DMSO: dimethyl-sulfoxide; CPA: *N*^6^-cyclopentyladenosine; *: comparison of the DMSO (five cycles) group with the three FSCPX-pretreated groups (the particular group is indicated by the color of the asterisk); ≠: comparison of the FSCPX (1 cycle) group with the FSCPX (5 cycles) group (comparison of the FSCPX (2 cycles) group to other FSCPX-pretreated groups did not provide significant results); the number of marks refers to the level of statistical significance (one mark: *p* < 0.05, two marks: *p* < 0.01, three marks: *p* < 0.001).

**Table 1 ijms-22-09831-t001:** Ectonucleotidases, enzymes participating in the extracellular adenosine formation [29].

Ectonucleotidases			
	ecto-nucleoside 5′-triphosphate diphosphohydrolase family (E-NTPDases) a.k.a. ecto-apyrases		
		group E-NTPDase1–4	
			**E-NTPDase1** a.k.a. **lymphocyte surface protein CD39** a.k.a. **ecto-apyrase CD39**
			E-NTPDase2 a.k.a. ecto-ATPase CD39L1
			E-NTPDase3 a.k.a. HB6
			E-NTPDase4: UDPase and LALP70
		group E-NTPDase5,6	
			E-NTPDase5 a.k.a. CD39L4
			E-NTPDase6 (?)
	ectonucleotide pyrophosphatase/phosphodiesterase family (E-NPP family) a.k.a. ecto-phosphodiesterase/pyrophosphatase family a.k.a. PC-1 family a.k.a. phosphodiesterase/nucleotide pyrophosphatase (PDNP) family		
		murine plasma cell differentiation antigen NPP1 (PC-1)	
		murine plasma cell differentiation antigen NPP2 (PD-Iα and autotaxin)	
		murine plasma cell differentiation antigen NPP3 (PD-Iβ a.k.a. B10 a.k.a. gp130^RB13-^^6^)	
	alkaline phosphatases a.k.a. non-specific ecto-phosphomonoesterases		
	**lymphocyte surface protein CD73** a.k.a. **ecto-5′-nucleotidase**		

Ectonucleotidases playing a key role in the interstitial adenosine production in the myocardium are marked in bold. Abbreviations (and the possible alternative splicing gene products) are in parenthesis. a.k.a.: also known as; (?): questionable role.

**Table 2 ijms-22-09831-t002:** Parameters (and their 95% confidence limits) of the Hill model (Equation (1)) fitted individually and globally to the averaged CPA (*N*^6^-cyclopentyladenosine) concentration–effect (E/c) data of four groups of rat as well as guinea pig left atria (presented in Figure 8).

		DMSO (5 Cycles)	FSCPX (1 Cycle)	FSCPX (2 Cycles)	FSCPX (5 Cycles)
R-i	E_max_ (%)	47.65	54.15	54.53	57.1
		(45.08 to 50.57)	(47.92 to 64.04)	(48.41 to 65.63)	(49.92 to 73.19)
	logEC_50_	−6.956	−5.558	−5.17	−5.123
		(−7.105 to −6.788)	(−5.809 to −5.209)	(−5.391 to −4.812)	(−5.346 to −4.689)
	n	0.797	0.739	0.68	0.815
		(0.632 to 1.046)	(0.547 to 1.006)	(0.533 to 0.852)	(0.567 to 1.136)
	r^2^	0.9474	0.9285	0.9526	0.9261
R-g	E_max_ (%)	49.99			
		(47.96 to 52.23)			
	logEC_50_	−6.879	−5.693	−5.311	−5.313
		(−7.019 to −6.73)	(−5.836 to −5.548)	(−5.45 to −5.173)	(−5.446 to −5.181)
	n	0.814			
		(0.719 to 0.926)			
	r^2^	0.9431	0.9261	0.9505	0.9161
GP-i	E_max_ (%)	93.41	93.92	95.78	96.2
		(89.19 to 97.92)	(89.71 to 98.72)	(90.73 to 101.9)	(91.4 to 101.9)
	logEC_50_	−7.319	−6.224	−6.028	−5.886
		(−7.457 to −7.18)	(−6.326 to −6.113)	(−6.148 to −5.891)	(−5.994 to −5.762)
	n	0.819	0.852	0.745	0.745
		(0.666 to 1.02)	(0.711 to 1.026)	(0.626 to 0.894)	(0.636 to 0.878)
	r^2^	0.9618	0.9829	0.9789	0.9812
GP-g	E_max_ (%)	94.35			
		(92.15 to 96.67)			
	logEC_50_	−7.303	−6.218	−6.052	−5.917
		(−7.398 to −7.209)	(−6.316 to −6.119)	(−6.146 to −5.958)	(−6.005 to −5.828)
	n	0.794			
		(0.727 to 0.869)			
	r^2^	0.9616	0.9826	0.9787	0.981

R-i: individual fit to rat E/c data; R-g: global fit to rat E/c data; GP-i: individual fit to guinea pig E/c data; GP-g: global fit to guinea pig E/c data; r^2^: coefficient of determination.

## Data Availability

The data presented in this study are available on request from the corresponding author.

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
