# Peer review of "A Body of Circumstantial Evidence for the Irreversible Ectonucleotidase Inhibitory Action of FSCPX, an Agent Known as a Selective Irreversible A1 Adenosine Receptor Antagonist So Far"

_ijms, 2021, doi:10.3390/ijms22189831_

Round 1
Reviewer 1 Report
This manuscript presented the research to describe "FSCPX paradox" identified through the previous research and tried to confirm that FSCPX was involved in the interstitial formation of adenosine. To do this, they observed changes in the negative intropic effect that occurred after pretreatment of ENT1 inhibitor NBTI or NBMPR and FSCPX together, and compared the effectiveness of FSCPX by processing both PSB-12379, which is CD73 inhibitor, and NBTI together.
However, it was generally difficult to understand because there was a lack of explanation for the reasons for this experimental design and a lack of explanation for the relationship between each inhibitors. Discussion also lacked interpretation of the results, and its meaning was unclear.
Page 2, line 58. Please write down the full name of CHA and CPA.
Page 3, Figure 1 legend: line 104: Please delete “)” of [10]).
Page 5, Secetion 2.2.1. line 178; Please explain each group of “all groups”
Page 6, Figure 3; Please explain the difference between “Control for NBMPR” and “Control for FSCPX & FSCPX+NBMPR”
Page 7, line 237-238: It is necessary to clarify what is the difference between “Control+NBTI+PSB (for Ado)” and “Control+PSB+NBTI (for Ado)”
Page 8, figure 5, left panel: “Graph for CPA” seemed unnecessary.
Page 8, figure 5, right panel: This graph seemed to have the same meaning as the right graph in figure 6.
Page 10, section 2.4.1: The subtitle is "before any in vitro treatment", but the content and the graph said that FSCPX had been pre-treated.
Page 14, line 477: “It may be speculated that the highly lipid-soluble FSCPX can more efficiently inhibit the membrane-bound CD73 than the highly water-soluble PSB-12379.” Figure 1 has already confirmed that FSCPX could not inhibit CD73. It is necessary to be clarified what it means.
Author Response
Response to Reviewer 1
Thank you for reviewing our manuscript. All your comments and suggestions have been addressed. Our answers are as follows (in sequence):
- Our experimental design (regarding the ex vivo investigations), the applied agents and the relationship among them were explained in two new paragraphs at the beginning of the Discussion. We hope that this insertion improves the level of interpretation of the results.
- The full name of CHA and CPA have been inserted into the Introduction.
- The unnecessary “)” has been deleted.
- It has been clarified what “all groups” means in the text objected (lines 183-184).
- The difference between groups “Control for NBMPR” and “Control for FSCPX & FSCPX+NBMPR” has been explained in the text belonging to the Figure 3 in the Results section (lines 193-195).
- The difference between groups “Control+NBTI+PSB (for Ado)” and “Control+PSB+NBTI (for Ado)” was also explained (lines 250-255).
- The left panel of Figure 5 presents the first adenosine E/c curve (generated in the absence of any adenosinergic agents) of all the groups participating in the given ex vivo investigation. As such, it demonstrates the homogeneity of atria used.
- The two blue and the two red E/c curves in the right panel of the Figure 5 are presented as one, pooled blue (Control curve) and one, pooled red E/c curve (PSB+NBTI curve) in the right panel of the Figure 6. In addition, the NBTI E/c curve and the PSB E/c curve are only seen in the right panel of the Figure 6. The similarity of right panels of Figures 5 and 6 stems from the same location of the Control and PSB E/c curves (that is per se an important observation).
- Terms “pretreatment” and “pretreated”, referring to the protocols using FSCPX and their outcome, only mean that FSCPX had already been washed out when the A1 receptor agonists were applied. In contrast, nitrobenzylthioinosines were present when the E/c curves were generated, thus terms “treatment” and “treated” were used concerning nitrobenzylthioinosines. So, the subtitle "before any in vitro treatment" is valid and meaningful.
- The Figure 2 (page 5) has confirmed that FSCPX could not inhibit CD39 and CD73 in an aqueous solution. In our work, we propose that it can inhibit them in an ex vivo tissue where a significant lipid compartment is present in the vicinity of CD39 and CD73.
Overall, the manuscript was reworked to improve its clarity. Thank You again for your work. We hope that you will find our revised manuscript suitable for publication in the IJMS.
Reviewer 2 Report
Gabor Viczjan et al. in manuscript entitled “A body of circumstantial evidence for the irreversible ectonucleotidase inhibitory action of FSCPX, an agent known as a selective irreversible A1 adenosine receptor antagonist so far” studied whether FSCPX blunted the cardiac interstitial adenosine accumulation in response to nucleoside transport blockade, by inhibiting CD39 and/or CD73. My major comment is that the founding about POM-1 as inhibitor of ecto-5‘-nucleotidase was showed previously; please see e.g. Kumar et al. SLAS Discovery 2020? Another major comment is the inhibiton assays with FSCPX that should be performed on the cells overexpressed with both eznzymes in the condition that would protect inhibitor from degradation. Such experiment should be added to the work before acceptation.
I have also a few minor comments:
- Please improve Fig.1 (the panels should not overlap).
- Standard deviation at Fig. 2 should be presented in different way to make it clear. Like in Fig. 1 panels should not overlap. Please correct all Figures that have more then 1 panel and in which legend covers the data.
- Results presented in Fig. 1 are highly confusing especially when compare with Fig. 4. It is not clear for me why not the same A1 agonist was used for both species?
- Figure 4 showed strong effect of FSCPX that not abolished the effect of NBMPR but rather make it negligible due to strength – please rewrite your conclusion to this results.
- The Fig. 5 – what is the different between Control+NBTI+PSB and Control+PSB+NTI?
- The statement that “Also in line with the results with CPA, this effect of PSB-12379 was weaker than that of FSCPX (cf. Figure 6, right panel and Figure 1, right panel).” (page 9) is not correct as PSB here add a little bit to the effect of NBTI not decrease it as it is in case with FSCPX.
Author Response
Response to Reviewer 2
Thank you for reviewing our manuscript. All your comments and suggestions have been addressed. Our answers are as follows (in sequence):
- The referenced article (DOI: 10.1177/2472555219893632) declares POM-1 and PSB-12379 as an inhibitor for CD39 and CD73, respectively, but does not state that POM-1 inhibits CD73 as well. To the best of our knowledge, this result of our present work has no antecedent in the literature.
- Choosing and using a cell-based enzyme inhibitor assay system to investigate CD39 and CD73 is a key issue for the continuation of our current research direction. However, due to its complexity, this investigation transcends the frames of the present manuscript and needs a new, separate project. The authors ask the Reviewer for a generous understanding of this issue.
- Presentation of all figures with two panels has been improved.
- Both panels of Figure 2 have vertically been elongated to make the error bars more visible. The error bars of some columns (e.g. negative controls) still remained undiscernible but this is due to the very small scatter in the related data sets.
- Comparing Figures 1 and 4 is really difficult due to complexity of the related experimental protocols. The reason for this complexity has been explained in the revised manuscript (lines 495-497).
- The recommended change has been carried out (lines 214-215).
- The difference between groups “Control+NBTI+PSB (for Ado)” and “Control+PSB+NBTI (for Ado)” was explained in the Results section (lines 250-255).
- To evaluate the effect of PSB-12379 and FSCPX on atria working in the presence of NBTI, the PSB+NBTI curve (Figure 6, right panel) and the FSCPX+NBTI curve (Figure 1, right panel), both ones represented by red diamond symbols, should be compared with their solely NBTI-treated counterpart, represented by blue triangles. Both curves with red diamonds exceed the corresponding curve with blue triangles. Moreover, the FSCPX+NBTI curve exceeds the NBTI curve to a considerably greater extent, as stated in the manuscript. The authors acknowledge that this would be more obvious if these two panels (right panels of Figure 6 and 1) could be placed side by side as a new figure, but it would be a double presentation and size limits do not allow this.
Taking all together, the manuscript has been reworked to improve its clarity and correctness. Thank You again for your work to facilitate this. We hope that you will find our revised manuscript suitable for publication in the IJMS.
Round 2
Reviewer 1 Report
The manuscript has been properly edited.
Reviewer 2 Report
Authors answered all the question raised in revision and accordingly improved MS. The work can be accepted in presented form.